# METTL14 regulates proliferation and differentiation of duck myoblasts through targeting MiR-133b

Qicheng Jiang[1], Tieshan Xu[2], Hailong Zhou[1], Zhepeng Xiao[3], Lingjing Xing[3], Xinli Zheng[3], Ping Yu[4], Zhe Chao[3], Zhongchun He[3], Wei Yang[5], Lihong Gu[3]*

1 School of Life and Health Sciences, Hainan University, Haikou, Hainan, China, 2 Tropical Crops Genetic Resources Institute, Chinese Academy of Tropical Agricultural Sciences, Haikou, Hainan, China, 3 Institute of Animal Science & Veterinary Medicine, Hainan Academy of Agricultural Sciences, Haikou, Hainan, China, 4 Institute of Tropical Agriculture and Forestry, Hainan University, Haikou, Hainan, China, 5 Qionghai Animal Husbandry and Veterinary Service Center, Qionghai, Hainan, China

* gulihong@hnaas.org.cn

## Abstract

The development of duck pectoral muscle has a significant impact on meat quality, and miRNA and m6A modification play key roles in this process. In the early stage, by using MeRIP-seq and miRNA-seq to analyze the pectoral muscle tissue of duck embryos at day 13 (E13), day 19 (E19), and day 27 (E27) of incubation, we found that METTL14, as a core component of the m6A methylation transferase complex, showed significant differences in expression at different developmental stages and may have an important impact on pectoral muscle development. In this study, qRT-PCR detection revealed that the expression of proliferation and differentiation marker genes CDK2, CyclinD1, MYOG and MYHC varied at different stages, with the highest m6A level at E13 and the lowest expression of METTL14 at the same stage. After constructing overexpression and interference vectors for METTL14, we found that METTL14 interference promoted the proliferation of duck embryo myoblasts and inhibited differentiation, while overexpression inhibited proliferation and accelerated differentiation. In particular, the overexpression of METTL14 increased the expression of miR-133b, whose precursor sequence contains m6A modification sites, suggesting that METTL14 may participate in the regulation of muscle development by affecting the expression of miR-133b. This study provides new insights into the molecular mechanisms of duck pectoral muscle development and offers potential molecular targets for the genetic improvement of duck pectoral muscle.

## 1. Introduction

m6A (N6-methyladenosine) is a common post-transcriptional RNA modification, which involves methylation at the sixth N position of adenosine and is widely present in various eukaryotic organisms [1]. m6A modification is an epigenetic change that occurs on RNA molecules, involving three main types of enzymes: writers, erasers, and readers. Writers contain specific conserved domains that are responsible for introducing m6A modifications on adenine residues in RNA. Correspondingly, erasers are responsible for removing this

**Data availability statement:** All relevant data are within the manuscript and its Supporting Information files.

**Funding:** This study was supported in part by National Natural Science Foundation of China General Program (31972553) and Chinese Modern Technology System of Agricultural Industry (CARS-42-50). The funders had no role in study design, data collection and analysis, decision to publish, or preparation of the manuscript.

**Competing interests:** The authors have declared that no competing interests exist.

modification, restoring the RNA to an unmodified state. The role of reader is to recognize m6A sites, thereby triggering various regulatory mechanisms, including RNA degradation and miRNA processing [2,3].

m6A modification plays a key regulatory role in cell differentiation and development. Studies have found that in stem cells, m6A modification can regulate the expression of key genes, affecting the balance of cell self-renewal and differentiation. For example, the knockout of the m6A modification factor METTL3 can cause an imbalance in the proliferation and differentiation of stem cells, leading to impaired stem cell function [4]. Other studies have shown that m6A modification is also involved in the directed differentiation of stem cells, regulating the fate determination of stem cells towards different cell lineages [5]. Secondly, m6A modification plays an important role in the process of embryonic development. During embryonic development, m6A modification plays a significant role in regulating the self-renewal and differentiation of embryonic stem cells, the early pattern formation of embryonic development, organ differentiation, and morphological development. For example, in the early embryonic development of mice, dynamic regulation of m6A modification on key genes has been found to affect the developmental direction and morphological formation of organs [6].

The processing of miRNA is influenced by m6A modifications. Pri-miRNA is the primary transcript of miRNA, ranging from hundreds to thousands of bases, containing one to several hairpin stem-loop structures, and possessing a 5' cap and a 3' poly(A) tail. Pri-miRNA is cleaved by the microprocessor complex Drosha-GDCR8 in the nucleus to form pre-miRNA with a single hairpin structure. Then, the Exportin-5-Ran-GTP complex transports pre-miRNA out of the nucleus into the cytoplasm, where the RNase Dicer enzyme, in combination with the double-stranded RNA-binding protein TRBP, cleaves the pre-miRNA into mature lengths, and finally, a single-stranded mature miRNA of about 22 bases in length is formed under the action of miRNA and the RNA-induced silencing complex [7]. Studies have found that the knockdown of m6A modification METTL3 affects m6A modification on pri-miRNA, thereby reducing the expression of mature miRNA, which has sparked a research surge in the regulation of pri-miRNA splicing by miRNA and RNA methylation [8].

METTL14 is a core enzyme in the m6A methylation modification system. METTL14 can form a stable heterodimer with METTL3 [2], where METTL3 provides catalytic activity, and METTL14 provides structural support and helps with substrate recognition. The m6A mark on mRNA targets by the METTL3/14 complex is bound by two families of reading proteins, namely proteins containing the YTH domain (YTHDF1-3 and YTHDC1-2) and insulin-like growth factor 2 mRNA-binding proteins (IGF2 BP1-3) [9,10]. m6A is one of the most common modifications of mRNA and regulates its stability and translation efficiency [11], so the METTL3/14 complex has also been found to have a regulatory role in various biological processes such as maintaining stem cell characteristics [12], tumor occurrence and development [13], gene expression regulation [14] and chromatin regulation [15]. In addition, METTL14 can also regulate organ or tissue proliferation and differentiation by targeting miRNAs [16]. Dong et al. Found METTL14 mediates m6a modification on osteogenic proliferation and differentiation of bone marrow mesenchymal stem cells by regulating the processing of pri-miR-873 [17].

The generation and development of skeletal muscle during the embryonic period are crucial biological processes in the growth axis of animals, which are essential for the normal growth and development of animals. Abnormal development of skeletal muscle often leads to embryonic muscle atrophy, deformity, or even death, so the study of its developmental regulatory mechanisms has important scientific significance and economic value [18]. Relevant reports have identified the growth and development laws of duck pectoral muscle, that is, on the 19th day of duck embryo incubation (E19), the pectoral muscle cells change

from proliferation to differentiation [19]. m6A methylation modification has widespread and conserved characteristics in eukaryotic organisms. miR-133b plays a crucial role in regulating skeletal muscle development by influencing the proliferation and differentiation of muscle stem cells and mitigating pathological changes such as fibrosis and inflammation [20]. Our research revealed significant differences in the expression levels of METTL14 and miR-133b, proliferation and differentiation of duck myoblasts across three embryonic stages. In light of this discovery, we decided to conduct further in-depth research on METTL14 to explore its specific role in the development of duck myoblasts [21,22]. The aim of this study is to investigate the roles of METTL14 and miR-133b in the development of duck embryonic skeletal muscle, particularly how they regulate the proliferation and differentiation of muscle cells through m6A RNA methylation. By constructing overexpression vectors and lentiviral vectors for METTL14, combined with qRT-PCR for quantitative gene expression analysis, and EDU detection to assess cell proliferation capacity, this experiment will delve into the functions and regulatory mechanisms of METTL14 and miR-133b during muscle development. The findings will contribute to understanding the impact of m6A modification on miRNA processing and the specific role of miR-133b in regulating the development of duck embryonic skeletal muscle. This will provide new insights into the molecular mechanisms of muscle growth and offer potential strategies for genetic improvement of muscle quality in the poultry industry.

## 2. Materials and methods

### 2.1 Ethics declarations

This experiment was performed in accordance with animal welfare principles and was conducted under protocols approved by the Chinese Universities Union for the Protection of Animals. All ducks were obtained from the Institute of Animal Science & Veterinary Medicine, Hainan Academy of Agricultural Sciences (IASVM-HAAS, Haikou, China). Ethical approval (reference number: IASVMHAAS-AE-202329) was conferred by the animal ethics committee of IASVM-HAAS, which is responsible for animal welfare. All the study protocol was conducted according to the arrive guidelines.

### 2.2 Sample collection

Duck embryos were obtained from Hainan Chuanwei Peking Duck Farming Co., Ltd., and eggs were selected on embryonic days 13 (E13), 19 (E19), and 27 (E27). The embryos were aseptically removed and their pectoral muscles were dissected; the pectoral muscles were placed into separate centrifuge tubes, immediately frozen in liquid nitrogen, and then stored at -80°C for further analysis.

### 2.3 Isolation, culture, and induction of differentiation of duck embryonic pectoral muscle myoblasts

After disinfecting the surface of 13-day-old duck eggs with alcohol, the duck embryos were removed and transferred to a cell culture dish containing 1% Penicillin Streptomycin in PBS to separate the pectoral muscles. The muscles were washed once with PBS containing Penicillin Streptomycin and placed in a new PBS dish with Penicillin Streptomycin. The muscles were then minced with scissors. The minced tissue was transferred to a centrifuge tube using DMEM without serum but containing Penicillin Streptomycin, the supernatant was discarded, and an appropriate amount of DMEM without serum but containing Penicillin Streptomycin was used to transfer the minced tissue to the centrifuge tube. Then, 0.25% trypsin was added and the mixture was digested at 37°C for 1 hour until the cells were completely dispersed. The digestion was stopped with DMEM containing 20% FBS. The mixture was passed through a

40 μm cell sieve, and the filtrate was centrifuged at 1,000 r/min for 10 minutes. The supernatant was discarded, and the cells were resuspended in DMEM containing 15% FBS. The cell count was adjusted to a concentration of $1 \times 10^6$ cells/mL, and the cells were aliquoted into cell culture flasks and cultured in a 37℃, 5%. When the cell density reached 80%, the medium was replaced with a differentiation medium, which consisted of DMEM supplemented with 2% horse serum and 1% penicillin-streptomycin.

### 2.4  Detection of genes and miRNAs related to the proliferation and differentiation phases of cells and tissues

PCR primers were designed for qRT-PCR based on the CDS sequences of duck CDK2, CyclinD1, MYOG, MyHC, and METTL14, using Primer Premier6.0 according to the GenBank database. β-actin was selected as the reference gene (Table S1 in File S1). For miRNA primers, a tailing method was used for design, and U6 was chosen as the reference gene (Table S2 in File S1).

### 2.5  qRT-PCR

For the qRT-PCR detection of proliferation and differentiation-related genes, the ChamQ Universal SYBR qPCR Master Mix Q711 kit was used, which employs the SYBR green dye method. Each sample was repeated three times. Data were analyzed using the $2^{-\Delta\Delta Ct}$ method.

CDK2 and CyclinD1 were chosen as proliferation marker genes, with β-actin serving as the reference gene. CDK2 is a gene that encodes a protein kinase, and the protein it encodes is a key regulator of the cell cycle [23]. CyclinD1 encodes a protein that is an important regulator of the cell cycle, involved in controlling critical biological processes such as the growth, division, and proliferation of cells [24].

MYOG and MYHC were selected as differentiation marker genes, with β-actin as the reference gene. MYOG is a key regulatory factor in the process of muscle differentiation and is a member of the muscle-specific transcription factor family. It can promote the transformation of muscle progenitor cells into mature muscle cells by regulating the transcription of muscle-specific genes. MYOG promotes the differentiation and maturation of muscle cells by regulating the expression of muscle-specific genes [25]. The MYHC gene encodes myosin heavy chain, which is an important component in the process of muscle cell fusion and muscle fiber formation [26].

### 2.6  Detection of m6A content in duck embryo RNA at different stages

The total m6A level in RNA samples from different developmental stages was determined using the P-9005 m6A RNA Methylation Quantification Kit (Colorimetric Method). First, RNA samples were prepared and processed according to the instructions of the kit. RNA samples and standard curve samples were mixed with RNA binding solution and added to the designated wells of a microplate. After incubation for a period, allowing the RNA to fully react with the binding solution and be immobilized on the plate, the plate was washed to remove unbound substances. A capture antibody was added to specifically bind to m6A-modified RNA. The plate was washed again to remove any non-specifically bound antibodies. A detection antibody was added to form an antibody-RNA complex. The plate was washed to remove excess detection antibody. A substrate solution was added to initiate the colorimetric reaction.

### 2.7  Design and synthesis of amplification primers

Primers MET1S and MET1AS were designed to amplify the duck METTL14 cDNA as a template. After the product was recovered, it was used as a template to add EcoR1 and BamH1

restriction enzyme sites, using primers MET2S and MET2AS for amplification. The specific primer sequences can be found in Table S3 in File S1.

## 2.8 Construction of overexpression vector

Amplify METTL14 from the extracted cDNA using the MET1S and MET1AS primers (Table S3 in File S1). Recover the obtained product using the Solarbio Agarose Gel DNA Recovery Kit. Use the DNA product as a template and perform PCR amplification with the MET2 primers, recover the amplified product. Digest the recovered DNA product with EcoR I and BamH I (TaKaRa) double enzymes, and then purify the double-digested product. Digest the pEGFP-N1 vector DNA with the same double enzymes and purify it. Use the TaKaRa DNA Ligation Kit to ligate the pEGFP-N1 plasmid vector and the double-digested DNA fragment. Extract the plasmid from the bacteria using the Solarbio Plasmid Mini Kit.

## 2.9 Transfection

In this study, Lipofectamine® 3000 is used as a transfection reagent. First, ensure that the cells have grown to a confluence of 70-90% in a 6-well plate. Then, dilute the required Lipofectamine® 3000 reagent with 250 μL of Opti-MEM® medium, and simultaneously dilute 5μg of DNA with an equal volume of Opti-MEM® medium to prepare a DNA premix. Next, add 7.5μL of P3000™ reagent to the DNA premix and mix thoroughly. Mix the diluted DNA premix with an equal volume of Lipofectamine® 3000 reagent and incubate at room temperature for 15 minutes to form a DNA-liposome complex. Finally, add 250μL of the complex to the cells to achieve efficient DNA transfection.

## 2.10 Construction of lentiviral interference vector

Utilize the online software BLOCK-iT™ RNAi Designer from ThermoFisher (http://rnaidesigner.thermofisher.com/rnaiexpress) to design shRNA interference sequences for METTL14. The sense strand sequence is in the order of 5' to 3' as follows: restriction site (BamHI), interference sequence, loop (TTCAAGAGA), reverse complementary sequence of the interference sequence, stop signal (TTTTTT), restriction site (EcoRI). The antisense strand sequence is in the order of 5' to 3' as follows: restriction site (EcoRI), complementary sequence of the stop signal (AAAAAA), interference sequence, complementary sequence of the loop (TCTCTTGAA), reverse complementary sequence of the interference sequence, restriction site (BamHI). Refer to Table S4 in File S1 for sequence details.

Select the pHBLV-U6-ZsGreen-Puro vector as the lentiviral interference vector, with packaging plasmids pLP1, pLP2, and pVSV-G, to form a four-plasmid lentiviral system. Digest the lentiviral interference vector with EcoRI and BamHI double enzymes. Anneal the synthesized shRNA sense and antisense nucleic acid sequences to form a double-stranded structure. Use the DNA Ligation Kit Mix to ligate the annealed shRNA with the linearized pHBLV-U6-ZsGreen-Puro.

## 2.11 Lentiviral packaging and collection

Revive low-passage 293T cells in advance and culture them using DMEM medium containing 10% FBS and 1% Penicillin Streptomycin. Revive and passage the cells three times. Seed 4 million cells onto a 10 cm cell culture dish and culture overnight. The next day, use the Lipofectamine® 3000 transfection reagent to co-transfect four plasmids, pHBLV-U6-ZsGreen-Puro, pLP1, pLP2, and pVSV-G, into the 293T cells. Use 4.3 μg of each plasmid, 41 μl of Lipofectamine® 3000, 35 μl of P3000, and 1.5 mL of Opti-MEM medium.

Replace the medium with fresh complete medium 6 hours after transfection. Collect the culture medium 24 hours and 52 hours after medium exchange. Centrifuge the collected virus-containing medium in a 15 mL centrifuge tube at 2,000 rpm for 5 minutes to remove cell debris and impurities. Then filter the supernatant through a 0.22 μm filter and add lentiviral concentrate to incubate overnight at 4°C. After centrifugation at 12,000 rpm for 10 minutes, discard the supernatant, resuspend the lentivirus in a small amount of DMEM, and store it at 4°C for future use.

### 2.12  Lentiviral infection of cells

Determine the titer of the stored lentiviral solution using the Lenti-Pac™ Lentiviral Titer Kit (Cat. Nos. LT005/ LT006). Then add the lentivirus to the duck embryo myoblast cell culture medium at a concentration of MOI = 50, change the medium after 24 hours, and observe under a fluorescence inverted microscope to check for fluorescence. After fluorescence is observed, replace the medium with complete medium containing 5 μg/mL of puromycin for drug screening (perform a puromycin killing curve test beforehand).

### 2.13  Cell proliferation detection

Use the kFluor488 Click-iT EDU Imaging Detection Kit for EDU detection. Seed $4 \times 10^4$ cells onto a 96-well plate one day in advance; the next day, incubate the cells with the EDU working solution for 2 hours. After incubation, remove the culture medium and add 50 μl of 4% neutral paraformaldehyde, and let it stand at room temperature for 30 minutes, then remove the fixative; add 50 μl of glycine solution (2 mg/mL) to each well and incubate at room temperature for 5 minutes; wash the cells twice with PBS containing 3% BSA; add 0.1 mL of 0.5% Triton X-100 in PBS and incubate at room temperature for 20 minutes. Prepare the Click-iT reaction mixture. Wash with PBS containing 3% BSA twice; add 0.1 mL of Click-iT reaction mixture to each well. Incubate at room temperature in the dark for 30 minutes, remove the reaction mixture, and wash twice. Then re-stain the DNA with Hoechst33342 diluted.

### 2.14  Statistical analysis of experimental data

Three biological replicates were used in this study, with each replicate measured three times. Data analysis and graph plotting were performed using GraphPad Prism 9.0 software (Graph-Pad Software, San Diego, CA, USA). For normally distributed data, one-way analysis of variance (ANOVA) was used for inter-group comparisons. For non-normally distributed data, the Kruskal-Wallis H test was used for non-parametric comparisons. Correlation analysis was conducted using Pearson's correlation coefficient. All data are presented as mean ± standard error of the mean (SEM), and a significance level of $P < 0.05$ was considered statistically significant.

## 3.  Results

### 3.1  The expression levels detection of proliferation and differentiation marker genes at different developmental stages in vivo

This study focused on the developmental process of duck embryos pectoral at different stages, selecting E13, E19, and E27 for the detection of proliferation and differentiation marker genes. CDK2 and CyclinD1 were chosen as proliferation marker genes, with β-actin serving as the reference gene. CDK2 (Cyclin-Dependent Kinase 2) is a gene encoding a protein kinase located on chromosome 16 of the duck genome. The protein it encodes is a critical regulator of the cell cycle. CyclinD1 (Cyclin D1) is a gene that encodes a protein, an essential cell

cycle regulator, involved in controlling key biological processes such as cell growth, division, and proliferation. qRT-PCR detection revealed that both CDK2 and CyclinD1 were highly expressed at E13 and gradually decreased, indicating that duck embryonic pectoral muscle cells were in a highly proliferative state at E13 (Fig 1A and 1B).

MYOG and MYHC were selected as differentiation marker genes, with β-actin as the reference gene. MYOG (Myogenin) is a key regulatory factor in muscle differentiation and belongs to the family of muscle-specific transcription factors. It promotes the conversion of muscle precursor cells into mature muscle cells by regulating the transcription of muscle-specific genes. MYOG fosters muscle cell differentiation and maturation through the regulation of these genes. The MyHC gene encodes myosin heavy chain, which is a crucial component in the process of muscle cell fusion and muscle fiber formation. qRT-PCR detection revealed that MYHC was highly expressed at E27, indicating that muscle cells were extensively fusing to form muscle fibers during this period. MYOG was highly expressed at E13, suggesting that this period may be the initial stage of differentiation for myoblasts (Fig 1C and 1D).

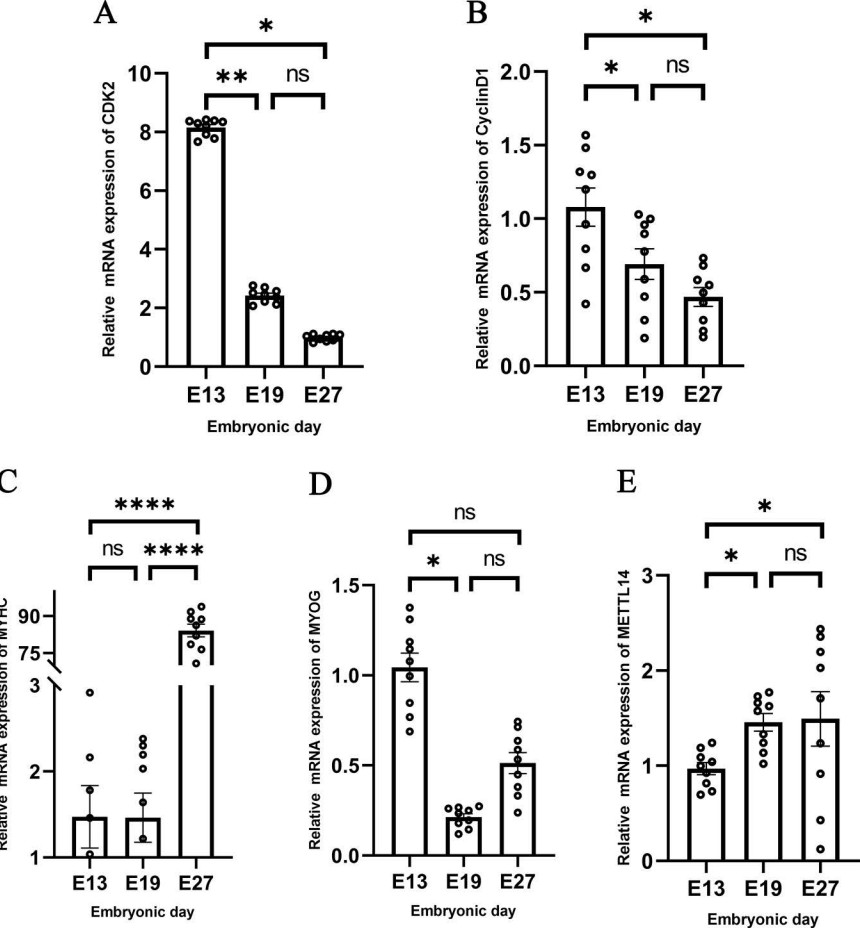

**Fig 1. Detection of genes at different stages in vivo. A** The expression of CDK2 at different stages. **B** The expression of CyclinD1 at different stages. **C** The expression of MYHC at different stages. **D** The expression of MYOG at different stages. **E** The expression level of METTL14 at different stages.

## 3.2  The expression levels detection of METTL14 at different developmental stages in vivo

For the analysis of and RNA-seq data from the E13, E19, and E27 stages [21], METTL14 was chosen as the subject for subsequent research. Therefore, this study further conducted qRT-PCR detection of METTL14 at the E13, E19, and E27 stages. The detection results are shown in the Fig 1E, where METTL14 has the lowest expression at the E13 stage, and the expression at E19 and E27 is almost the same but higher than that at the E13 stage.

## 3.3  The expression levels detection of muscle specific miRNAs at different developmental stages in vivo

In this study, we selected U6 as the internal reference gene and, based on the sequencing data from previous studies, screened and validated six miRNAs related to muscle development using qRT-PCR. These miRNAs include: bta-miR-133b_R-1 [27], mdo-miR-26-5p_R+1 [28], mmu-miR-135a-5p [29], eca-miR-206 [30], has-miR-133a-5p [31], and has-miR-148a-3p_R-2 [32] (Table S5 in File S2). The qRT-PCR results were consistent with the sequencing data, showing variations in the expression levels of these miRNAs at different developmental stages, suggesting their potential regulatory roles in muscle development (Fig 2A–2F).

## 3.4  In vivo detection of m6A at different stages

In previous studies, we observed that a number of genes exhibited substantial changes in m6A modification levels across three developmental stages. However, the overall m6A level across the three stages was not investigated. Therefore, this study further employed the m6A RNA Methylation Quantitation Kit P-9005 to explore the overall m6A levels in the RNA from the three stages. The results showed that the percentage of m6A in the total RNA of the pectoral muscle tissue at the three developmental time points, E13, E19, and E27, were 0.5249%, 0.4870%, and 0.4830%, respectively (Fig 2G). The data indicate that among the three developmental time points of pectoral muscle, the methylation expression level is highest at E13, while the methylation levels at E19 and E27 were consistent with each other. To build upon these in vivo findings, we next sought to validate the role of m6A modification in muscle development through in vitro experiments.

## 3.5  Cell culture and detection of proliferation and differentiation

The isolated duck embryo myoblasts were cultured in a complete medium composed of DMEM supplemented with 15% FBS and 1% Penicillin Streptomycin, which allowed the cells to grow normally (Fig 3A and 3B). When the cell density reached 80%, the old medium was discarded, and the cells were washed with PBS. Then, a differentiation medium was used to replace the old one to induce differentiation of the myoblasts, with the medium being changed daily. Upon successful induction of differentiation, myotubes, which are thick tubular structures, can be observed, and multinucleated cells appeared (Fig 3C and 3D).

## 3.6  Detection of cell differentiation marker genes

Cell fusion phenomena began to appear 1 day after the induction of differentiation, and by the 2nd day, the fusion of myotubes could be clearly observed. It took a week for the cells to fully fuse into myotubes and begin to detach, with no distinct boundaries between different stages of differentiation. Therefore, this study detected of differentiation marker genes at different time points (1 day, 2 days, and 4 days) after the induction of differentiation. The results are shown in Fig 3E and 3F, where both MYOG and MYHC expression levels gradually increased with the duration of induced differentiation. It can be inferred from this that within the first 4

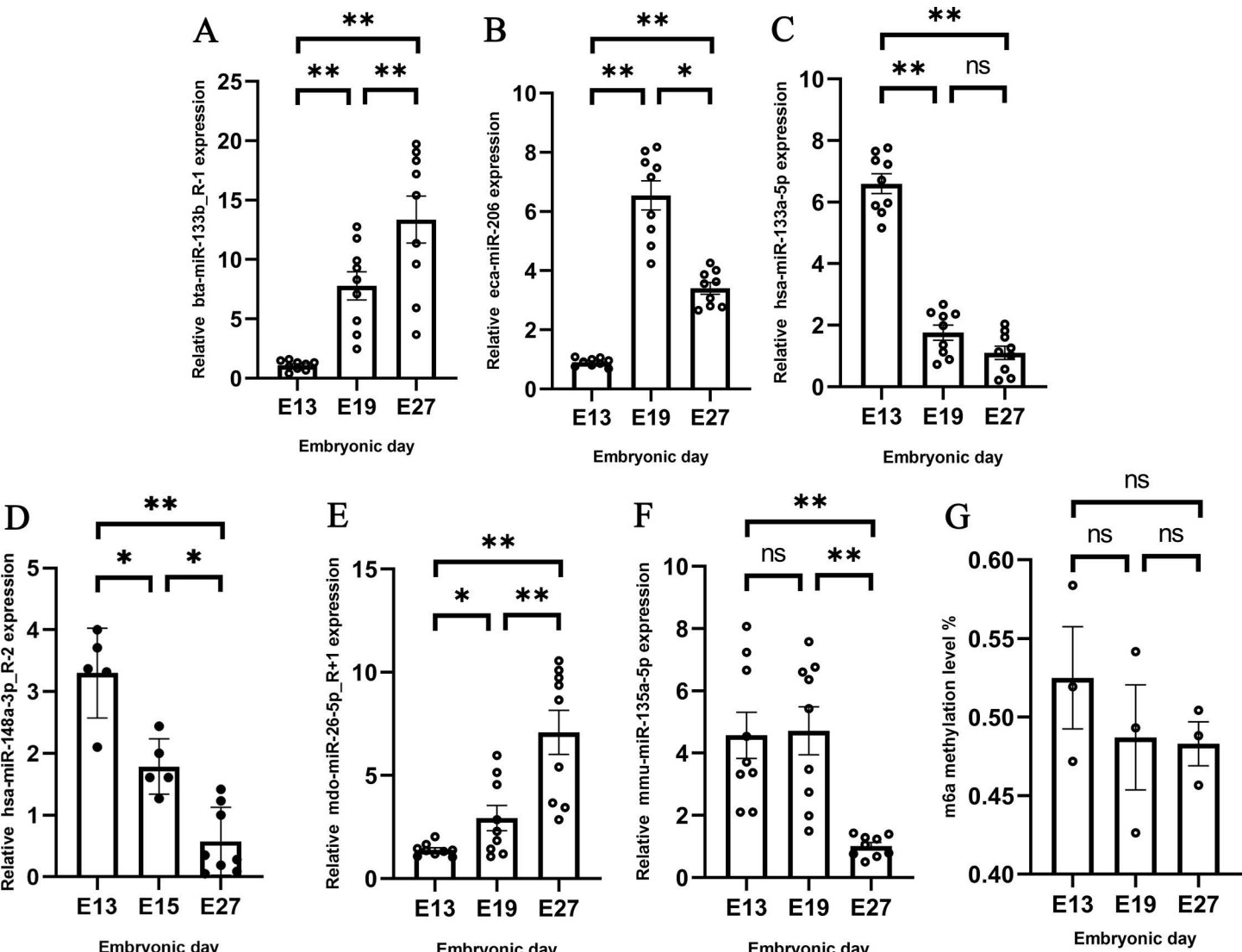

**Fig 2. Detection of miRNAs and m6A level at different stages in vivo. A** The expression of bta-miR-133b-R-1 at different stages. **B** The expression of eca-miR-206 at different stages. **C** The expression of mmu miR-135a-5p at different stages. **D** The expression of has-miR-148a-3p-R-2 at different stages. **E** The expression of bta-miR-26a R + 3 at different stages. **F** Expression of has miR-133a-5p at different stages. **G** The content of m6A at different stages.

days of induced differentiation, the expression levels of MYOG and MYHC rise as the degree of differentiation increases.

## 3.7 The effect of METTL14 overexpression and down-expression on myoblast proliferation and differentiation

qRT-PCR was used to detect the expression of METTL14 after transfection, and the results, as shown in Fig 4A, indicate a significant increase in METTL14 expression. The concentrated lentivirus was used to infect cells at an MOI of 50, and the result showed a significant reduction in METTL14 expression after interference (Fig 4B).

The EDU (5-Ethynyl-2'-Deoxyuridine) staining method allows for a direct observation of cell proliferation. In this study, EDU staining was performed on cells with METTL14

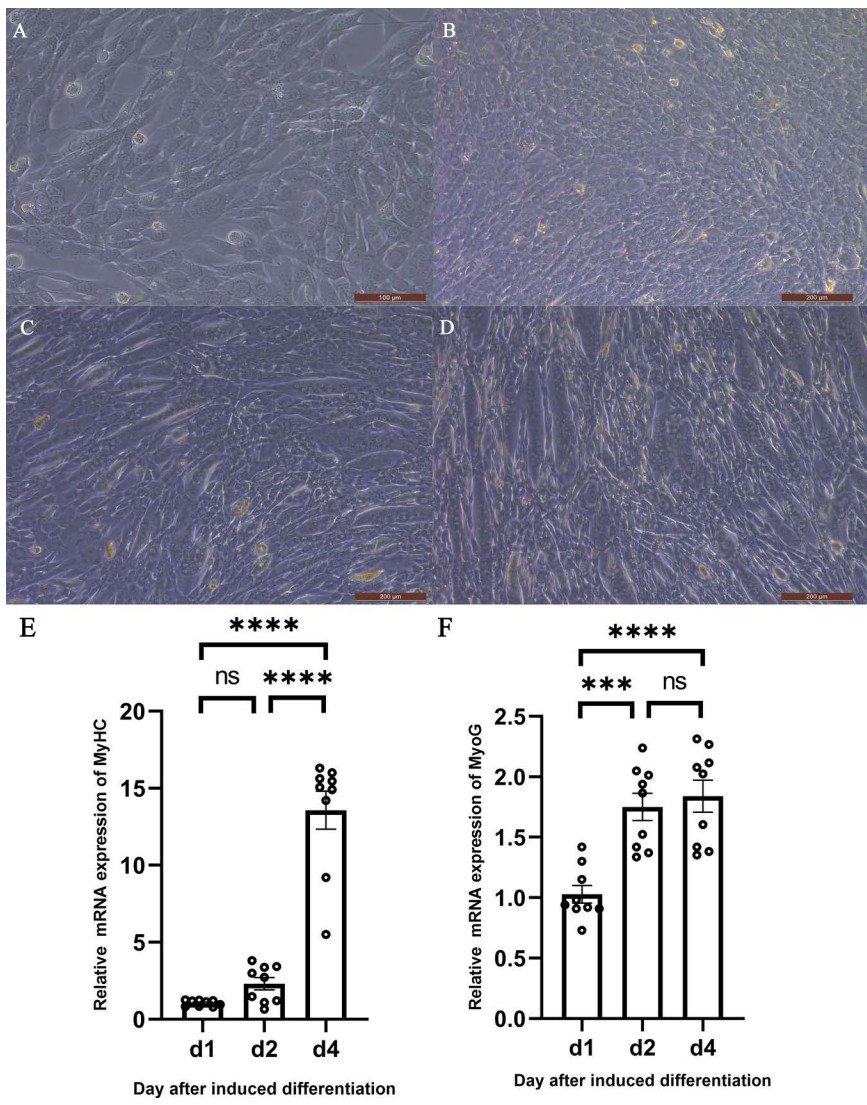

**Fig 3. Detection of cell proliferation and differentiation culture. A and B** Duck embryo myoblasts were successfully isolated and grew normally. **C and D** Induced differentiation of duck embryonic myoblasts results in cell fusion and myotubes. **E** Expression of MYHC at 1, 2, and 4 days after induction of differentiation. **F** Expression of MYOG at 1, 2, and 4 days after induction of differentiation；d1, d2, and d4 are days 1, 2, and 4 after induction of differentiation.

interference and overexpression, and the results are shown in Fig 4C. Analyzing the data, the EDU-positive cell count for the shMETTL14 group was 44.40%, for the MOCK group it was 29.61%, and for the pE-METTL14 group, it was only 11.23% (Fig 4D). It can be observed that when METTL14 is knocked down, the cell's proliferative capacity is enhanced, whereas overexpression of METTL14 leads to a decrease in cell proliferation. Therefore, it can be concluded that the knockdown of METTL14 can promote the proliferation of duck embryo myoblasts, while the overexpression of METTL14 inhibits their proliferation rate.

To explore the impact of METTL14 on the differentiation of duck embryo myoblasts, interference and overexpression of METTL14 were performed on the cells, followed by the detection of differentiation marker genes MYOG and MYHC on the second day of induced

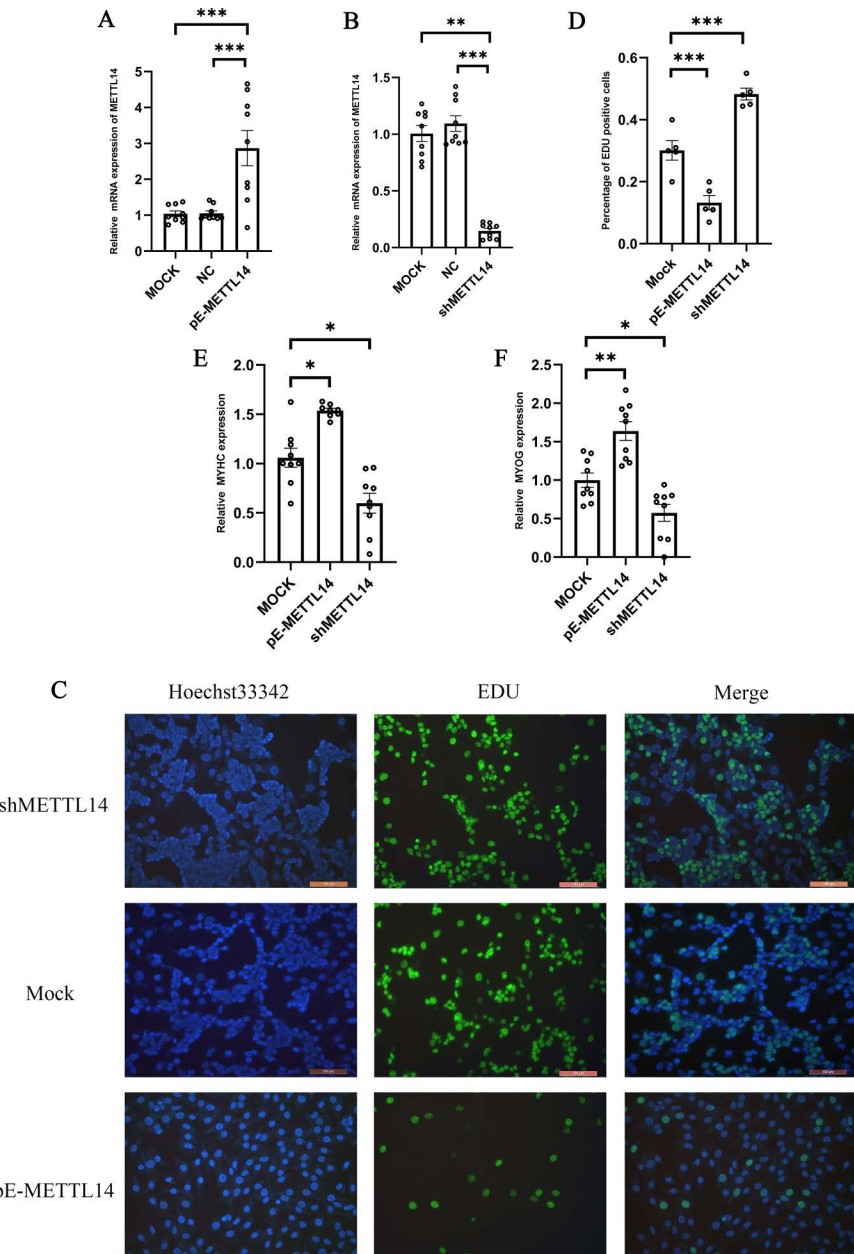

**Fig 4. The METTL14's impact on cell proliferation and differentiation. A** The expression level of METTL14. **B** The expression level of METTL14; MOCK represents the unprocessed group; NC is a negative control; pE-METTL14 is overexpressed in the METTL14 group; shMETTL14 is the interference METTL14 group. **C** EDU detection; Hoechst44442 represents nuclear staining, EDU represents EDU positive cells, and merge represents fusion in the previous image; shMETTL14 is the interference METTL14 group; MOCK is an unprocessed group; pE-METTL14 is overexpressed in the METTL14 group. **D** The proportion of EDU positive cells. **E** The expression of MYHC. **F** The expression of MYOG; shMETTL14 is the interference METTL14 group; MOCK is the control group; pE-METTL14 is overexpressed in the METTL14 group.

differentiation. The results showed that the expression levels of both MYOG and MYHC were increased after the overexpression of METTL14, while the expression levels of both MYOG and MYHC were decreased after the interference of METTL14. This indicates that the

interference of METTL14 inhibits the rate of cell differentiation, whereas the overexpression of METTL14 can enhance the rate of cell differentiation(Fig 4E and 4F).

### 3.8 METTL14's impact on miR133b

Reports in existing literature indicate that METTL14 can regulate the precursor sequence of miRNA, known as Pri-miRNA, and subsequently influence the expression of miRNA. Additionally, m6A modification is predominantly found on the RRACH motif. Since the majority of Pri-miRNA sequences remain unexplored, but the sequences of the downstream Pre-miRNA are known, this study predicts the sequence 500 base pairs upstream and downstream of Pre-miRNA as the Pri-miRNA sequence.

By comparing the RRACH motif, it was discovered that there is a significant presence of motifs in miR-133b that match the m6A modification site (Fig 5A). Consequently, this study proceeded to detect miR-133b in cells after METTL14 interference and overexpression. The results showed that the expression level of miR-133b increased with the overexpression of METTL14, and decreased with the interference of METTL14 (Fig 5B). This suggests that changes in METTL14 can affect the processing of miR-133b, which might be due to METTL14 acting on the motif within the precursor sequence of miR-133b, thereby influencing the maturation of miR-133b. Of course, this still requires substantial further research.

## 4. Discussion

METTL14 is a key RNA modification enzyme involved in the regulation of m6A RNA methylation modification [33]. METTL14 binds to RNA through its RNA-binding domain, thus locating on the target RNA, and interacts with METTL3 to form an active complex, catalyzing the formation of m6A modification [34]. METTL14 has been proven to be involved in various biological processes, including the regulation of embryonic development [35], stem cell fate determination [36], cell proliferation [37] and differentiation [38], among other key biological processes. This study found that overexpression of METTL14 can inhibit the proliferative

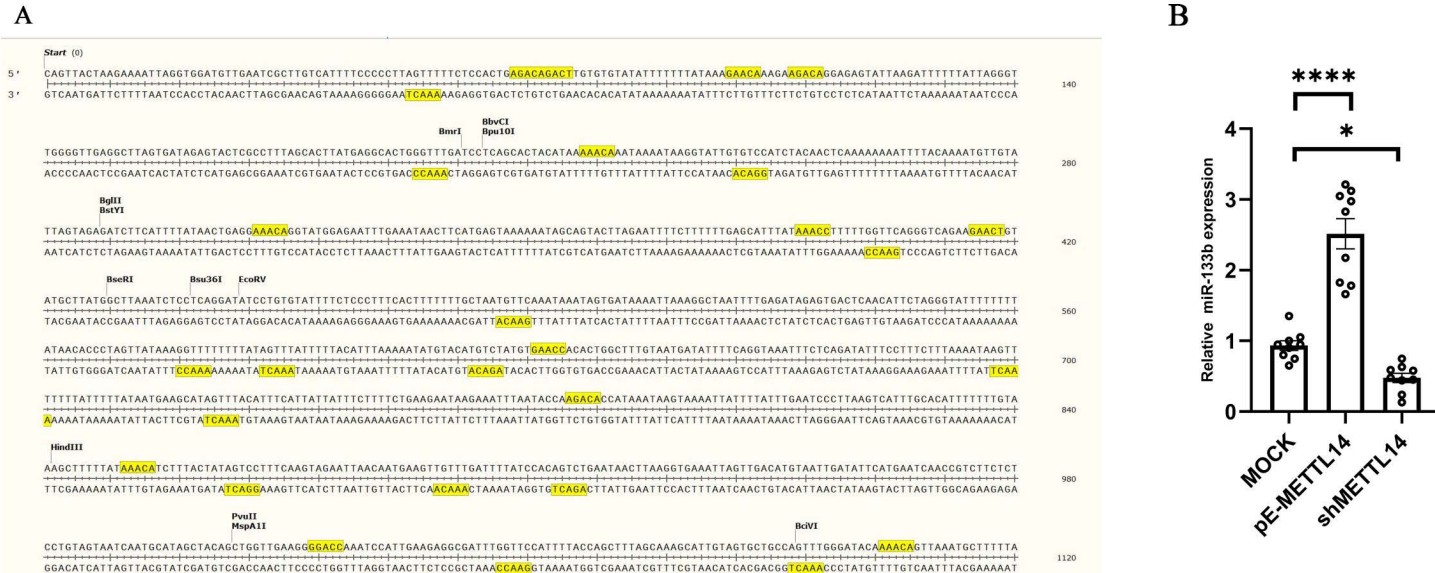

**Fig 5. The METTL14's impact on miRNA. A** Motifs in the pri-miR-133b. **B** MiR-133b expression level; shMETTL14 is the interference METTL14 group; MOCK is the control group; pE-METTL14 is overexpressed in the METTL14 group.

capacity of myoblasts, accelerate the level of myoblast differentiation, and promote the expression of miR-133b. This reveals the existence of a regulatory relationship between METTL14 and miR-133b.

The incubation period for duck embryos typically lasts for 30 days. From the start of incubation until the 5th day, this phase is primarily characterized by the development of various tissues and organs within the duck embryo, with muscle tissue beginning to form from the mesoderm. Between days 6 and 14 of incubation, the embryo starts to exhibit distinct morphological features, such as the outlines of wings and legs. From day 15 to day 21, the organs of the embryo have essentially formed and begin to enter a stage of growth and maturation. Finally, from day 22 to day 28, the embryo's development is nearly complete, and it starts to absorb the remaining yolk sac, preparing to hatch. Gu et al. found morphological analysis of duck pectoral muscle indicates that E19 or E20 is a transitional point for the pectoral muscle of Beijing ducks (from proliferation to fusion). The expression patterns of MRF4, MyoG, and MSTN suggest that E19 or E20 is the point of fastest development of the pectoral muscle, and E19 or E20 is a critical turning point in the development of pectoral muscle in embryonic Beijing ducks [19]. The results of this study found that CDK2 and CyclinD1 are highly expressed at E13 and gradually decrease. This indicates that duck embryo pectoral muscle cells are in a highly proliferative process at E13. MYHC is highly expressed at E27, indicating that muscle cells are extensively fusing to form myofibers at this time. MYOG is highly expressed at E13, suggesting that this point may be the period when myoblasts enter the early stages of differentiation (Fig 1).

The mature study of m6A began in 2011, that is, the identification of the first FTO of RNA m6A, the enrichment of antibodies, the development of high-throughput sequencing technology, and the precise mapping of the m6A distribution at the transcriptome level could be achieved [39]. m6A has a wide range and is conserved; these m6A modifications have a conserved modification motif RRACH (R represents A or G, H represents A, U, or C), and are enriched in the promoter region, stop codon region, and specific motifs of mRNA. m6A modification selectively forms at the adenine (A) site of the conserved sequence RRACH in mRNA [40]. Interestingly, this study found that the predicted precursor sequence of miR-133b contains multiple motifs that match RRACH, and these motifs may be the binding sites for METTL14(Fig 5). This will provide ideas for a deeper understanding of the targeting relationship between METTL14 and the precursor sequence of miR-133b.

The report that the m6A modification system can regulate miRNA processing was first in 2015, when Claudio R et al [8]. published a paper titled "m6A regulates pri-miRNA recognition processing" in Nature, detailing how METTL3 participates in m6A modification on pri-miRNA, promoting the recognition of pri-miRNA by DGCR8, thereby promoting the maturation of microRNA, and the study's conclusions is conserved in this system. In the same year, Claudio R et al. published another paper titled "m6A recognition enzyme HNRNPA2B1 mediates RNA processing" in Cell, again proving that m6A on pri-miRNA is recognized by HNRNPA2B1, which then activates the pre-miRNA processing pathway, subsequently affecting the processing of the mature body [41]. This study found that METTL14 has significant differences in the development process of duck embryo pectoral muscle and subsequently verified METTL14. It was found that METTL14 can regulate the expression of miR-133b, and the precursor sequence of miR-133b has m6A modification sites, so there may be a pathway where METTL14 mediates m6A RNA methylation to regulate miRNA processing and affect the proliferation and differentiation of duck myoblasts. Similarly, it has been found that in primary hepatocellular carcinoma cells, the maturation of miR-126 is mediated by the methylation transferase METTL14, and a decrease in the expression of mature miR-126 can cause the transfer of primary hepatocellular carcinoma

cells [42]; when METTL3 is knocked down, the expression of mature miRNA is reduced due to the impact on m6A modification on pri-miRNA, further confirming the regulation of pri-miRNA processing by m6A modification.

Currently, many miRNAs lack primary miRNA (pri-miRNA) sequence information in databases such as miRBase and MicroRNAdb. Pri-miRNAs are present in the body for an extremely short duration, making their detection by conventional PCR challenging. The current methods for obtaining pri-miRNA sequences primarily involve RACE experiments, where primers are designed based on known precursor miRNA (pre-miRNA) sequences to amplify the full-length sequence [43], or the use of specific high-precision detection kits, which unfortunately have a relatively high failure rate. Therefore, this study proposes predicting the pri-miRNA sequence by taking the position of pre-miRNA in the genome and extending 500 base pairs before and after it (Fig 5A). Existing research has shown that the transcription of pri-miRNA is regulated by various factors, including transcription factors and epigenetic modifications. m6A is one such epigenetic modification that targets adenosine within specific RRACH motifs. This study identified a significant number of RRACH motifs in pri-miRNA-133b, leading to the hypothesis that the transcription of pri-miRNA-133b may be subject to m6A modification.

miRNAs are a class of short non-coding RNA, approximately 21-25 nucleotides in length, that are widely present in eukaryotic organisms and participate in the regulation of gene expression. miRNAs regulate the stability and translation of these mRNAs by binding to target genes, thereby affecting the expression levels of the target genes [44]. As an important regulatory factor, miRNA plays a key role in developmental regulation, participating in important processes such as embryonic development [45], Organogenesis [46], and tissue differentiation [47]. The role of miRNA in the developmental process is multifaceted. Firstly, in the early stages of embryonic development, miRNAs are involved in the differentiation and directed development of embryonic stem cells. This study first proved that METTL14 plays an important role in the proliferation and differentiation processes of duck myoblasts, but it did not prove that METTL14 is directly involved in the proliferation and differentiation process of duck myoblasts. Therefore, in order to find potential downstream regulatory factors, miRNA-seq was performed in the previous stage, and some miRNAs related to muscle development were found, including chi-miR-148a-5p_1ss10AT, hsa-miR-133a-5p, mdo-miR-29b-2-5p_L-1R+1, mmu-miR-29b-3p_R-1, bta-miR-29c, mmu-miR-135a-5p, hsa-miR-148a-3p_R-2, eca-miR-206, mdo-miR-26-5p_R+1, hsa-miR-27b-3p, bta-miR-133b_R-1. Further validation is required to determine which of these miRNAs are regulated by METTL14. From this, it was found that the expression of miRNA-133b is affected by METTL14. And miRNA-133b has been proven to be involved in the proliferation [48] and differentiation [49] of myoblasts. From this, a potential METTL14/miRNA-133b regulatory pathway was discovered (Fig 6).

## 5. Conclusion

This article showed that both CDK2 and CyclinD1 were highly expressed at E13 and gradually decreased, MYHC was highly expressed at E27, and MYOG was highly expressed at E13. The expression levels of six miRNAs related to muscle development vary at different stages. Further verification results revealed that the knockdown of METTL14 can promote the proliferation of duck embryo myoblasts and inhibit the rate of cell differentiation, while the overexpression of METTL14 inhibits the proliferation rate and increases the differentiation rate. After the overexpression of METTL14, the expression of muscle development-related miRNA, miR-133b, increased; further prediction of the miR-133b precursor sequence revealed that the

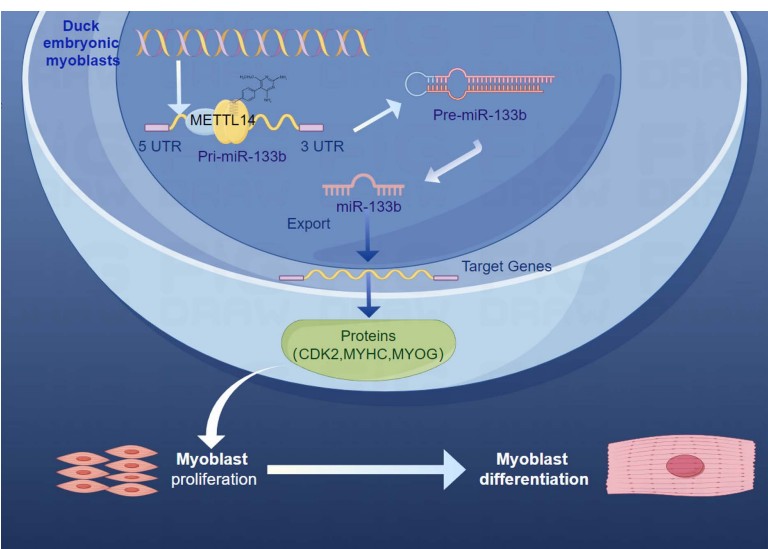

**Fig 6. Graphical abstract of METTL14 regulates proliferation and differentiation of duck myoblasts through targeting MiR-133b.**

precursor sequence contains multiple motifs that match the m6A modification site. Therefore, it is speculated that METTL14 affects the processing of miR-133b, leading to further effects on the proliferation and differentiation of duck embryo myoblasts.

## Supporting information

**File S1. Primer sequence.**
(DOCX)

**File S2. miRNAs related to skeletal muscle development.**
(DOCX)

## Author contributions

**Data curation:** Qicheng Jiang, Lingjing Xing.

**Formal analysis:** Zhepeng Xiao.

**Funding acquisition:** Lihong Gu.

**Methodology:** Tieshan Xu.

**Resources:** Xinli Zheng, Ping Yu, Zhe Chao, Zhongchun He, Wei Yang.

**Supervision:** Hailong Zhou.

**Writing – original draft:** Qicheng Jiang.

**Writing – review & editing:** Qicheng Jiang, Lihong Gu.

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
