## [Decision Letter · Decision Letter 0]

6 Jan 2025

PONE-D-24-38664METTL14 Regulates Proliferation and Differentiation of Duck Myoblasts Through Targeting MiR-133bPLOS ONE

Dear Dr. Gu,

Thank you for submitting your manuscript to PLOS ONE. After careful consideration, we feel that it has merit but does not fully meet PLOS ONE’s publication criteria as it currently stands. Therefore, we invite you to submit a revised version of the manuscript that addresses the points raised during the review process.

We look forward to receiving your revised manuscript.

Kind regards,

Abdul Qadir Syed, PhD

Academic Editor

PLOS ONE

Journal Requirements:

Reviewers' comments:

Reviewer's Responses to Questions

**Comments to the Author**

1. Is the manuscript technically sound, and do the data support the conclusions?

Reviewer #1: Partly

Reviewer #2: Yes

2. Has the statistical analysis been performed appropriately and rigorously? 

Reviewer #1: No

Reviewer #2: Yes

3. Have the authors made all data underlying the findings in their manuscript fully available?

Reviewer #1: Yes

Reviewer #2: Yes

4. Is the manuscript presented in an intelligible fashion and written in standard English?

Reviewer #1: Yes

Reviewer #2: Yes

5. Review Comments to the Author

Reviewer #1: In this study the authors establish a role for METTL14 and its putative target miR-133b in the transition from proliferative tissue establishment to differentiation, with regards to the size and function of the duck non-model organism pectoral muscle. The authors use changes in the expression of proliferative and differentiation markers to establish the transitional point of interest, embryonic day 19, and establish days E13 and E27 as proliferative or differentiating, respectively (Fig. 1, 3). Indeed, the authors were also able to overlay changes in METTL14 expression to this transition period, correlating METTL14 expression with the change in proliferative to differentiation gene expression. They subsequently demonstrate that METTL14 drives this change via overexpression and RNAi (Fig 4). Finally, the authors show that expression of miR-133b, a miRNA previously demonstrated to regulate proliferation of myoblasts in mammalian systems, can be increased by overexpression of METTL14. This is supported by the presence of several potential METTL14 regulatory signature sequences in the sequence surrounding miR-133b (Fig 2, 5).

Some of these relationships have already been demonstrated in model organisms, most notably the role for both METTL3/METTL14 complex and miR-133b regulating tissue differentiation/proliferation, though in independent studies. However, the linking of miR-133b as a substrate of the METTL 3/14 complex, establishing a mechanism for this known developmental programming switch, and ex vivo evaluation of the above in a non-model system are all novel. The biggest weakness of the paper is the establishment of miR-133b. The results as presented certainly demonstrate a relationship between METTL14 and miR133b, and correlate miR-133b expression with tissue proliferation, but a functional or causative relationship between miR-133b and the proliferative status of tissue/genes of interest was not investigated. Establishing a specific mechanism for METTL14-mediated transition to musculature differentiation, particularly in a novel organism and with novel target miR-133b would be an interesting additional exploration.

Detailed comments

Major:

1. METTL14 and miR-133b do have putatively relevant function and the authors demonstrate correlated expression levels, but this is not functionally or causatively investigated with regard to the role of miR-133b muscle cell proliferation. A phenotype for overexpressing or inhibiting the regulatory activity of miR-133b, without the confounding factor of METTL14 over/underexpression, would much more strongly support the title and some conclusions of the manuscript. Could the authors transfect synthetic/isolated miR-133b utilizing their already established lentiviral or lipofectamine system?

2. Genes assessed in figure 1 are used as markers for developmental stage and/or evidence of cellular processes without clarification or discussion (example – line 285/290). Elaborating on each slightly would much more strongly support their role in the manuscript, particularly those not assessed in the reference establishing E19 as a developmental transition - CDK2, CyclinD1, and MYHC.

3. In lines 307-316 it’s unclear how or why these six particular miRNAs were selected for analysis. The text mentions previous studies, but none are cited.

4. A few statistical analyses/pieces of information are not readily visible on figures. Differences in m6a methylation are described as though significant, but significance is not shown on Fig2G. Figures 3 and 4D similarly appear to show differences in differentiation gene expression and proliferative rate respectively, but the results of statistical analysis are not shown. Lastly, it appears as though the highlighted regions of figure 5A are METTL14 m6a modifying sequences, but this is not mentioned in the legend/text and the sequences in the figure are somewhat difficult to read.

5. Similarly, I was not able to determine sample sizes throughout.

Minor

1. The acronym ‘m6a’ is not formally defined.

2. A few conjugation/sp/grammatical errors, e.g.: line 119 “our previously research”, the timeline of sample collection on line 139 is somewhat unclear in “eggs were selected… post-hatch” (does this mean individuals?), line 142 “stored at -80 for further.”, line 318 “In the previous research” should perhaps be “in previous studies”, line 349 “conducted the detection” should just be “detected”, line 471 “apply to all animals” should perhaps be “is conserved in this system” to maintain scope, the sentence “And want to further verify..” on line 519-520 is an incomplete sentence.

3. Some reorganizations could help significantly with regards to reader comprehension as well as supporting the authors’ argument. While the manuscript as presented does follow a logical ordering, moving from in vivo proof of concept to more causative in vitro experiments, some sections feel out of place. Most notably sections 3.5 and 3.6 disrupt the ideological story by pivoting rapidly from m6a modified RNAs in vivo, back to the developmental status marker genes in sections 3.1 and 3.2, and then return to the change in expression of specific miRNAs in the newly established in vitro system.

4. The differentiation media used to establish the ex vivo system in line 340 is not specified in the materials and methods or cited in references.

5. The first few sentences of 3.7 are redundant with the well described methods section. Of more use here would be the contents of and purpose for the plasmids transfected.

6. In lines 375-377 the conclusions of the proliferation assessment are offered before the data. This sentence should instead be centered on the purpose and hypothesis for these data. In addition, the proliferative markers assessed in lines 382-390 are more consistent with the preliminary experiments overall. Flipping these two sets of data would allow for increasingly granular assessment of proliferation.

7. Some information that, in the introduction or alongside data, would support the relationship between METTL14 and miR-133b is instead in the discussion. E.g. the discussion in lines 445-447 establishing E19 as a critical turning point in duck pectoral muscle, or the established regulatory role for miR-133b in differentiation/proliferation of other muscle tissue in other systems in 521-523.

8. There are a few places when summarizing data in the discussion, e.g. references to the expression of MYOG and MYHC in 450-453, that would benefit from directing the reader back to the relevant figure (here, fig. 1).

9. In lines 481-486, the authors present a proof-of-concept for the METTL3/14 complex in RNA regulation, but it is unclear the extent to which each METTL protein is acting. Is there a reason the authors focused only on METTL14?

Reviewer #2: In the current manuscript, “METTL14 Regulates Proliferation and Differentiation of Duck Myoblasts Through Targeting MiR-133b” the authors have shown the importance of METTL14, which is a core component of the m6A methylation transferase complex, in the development of Duck pectoral muscle. Further, authors have also shown that the overexpression of METTL14 increased the expression of miR-133b, whose precursor sequence contains m6A modification sites, suggesting that METTL14 may participate in the regulation of muscle development by affecting the expression of miR-133b. Overall this study provides a new perspective of the molecular mechanisms involved in the development of duck pectoral muscle. This research appears to have therapeutic significance since it offers potential molecular targets for the genetic improvement of duck pectoral muscle. I recommend the manuscript for publication subject to responses to the following comments

Minor comments

1) Authors should consider adding individual data points in all of the bar graphs presented in result sections.

2) Authors should consider making a graphical abstract to summarise their findings.

3) Authors are claiming that the overexpression of METTL14 increased the expression of miR-133b, whose precursor sequence contains m6A modification sites, suggesting that METTL14 may participate in the regulation of muscle development by affecting the expression of miR-133b. It would be interesting if authors could also show the effect of miR-133b mimic and inhibitor on pectoral muscle development and its effect on expression of METTL4.

6. PLOS authors have the option to publish the peer review history of their article (what does this mean? ). If published, this will include your full peer review and any attached files.

**Do you want your identity to be public for this peer review?** For information about this choice, including consent withdrawal, please see our Privacy Policy .

Reviewer #1: **Yes: ** Joshua Thompson

Reviewer #2: No

---

## [Author Response · Author response to Decision Letter 1]

21 Jan 2025

Reviewer #1: In this study the authors establish a role for METTL14 and its putative target miR-133b in the transition from proliferative tissue establishment to differentiation, with regards to the size and function of the duck non-model organism pectoral muscle. The authors use changes in the expression of proliferative and differentiation markers to establish the transitional point of interest, embryonic day 19, and establish days E13 and E27 as proliferative or differentiating, respectively (Fig. 1, 3). Indeed, the authors were also able to overlay changes in METTL14 expression to this transition period, correlating METTL14 expression with the change in proliferative to differentiation gene expression. They subsequently demonstrate that METTL14 drives this change via overexpression and RNAi (Fig 4). Finally, the authors show that expression of miR-133b, a miRNA previously demonstrated to regulate proliferation of myoblasts in mammalian systems, can be increased by overexpression of METTL14. This is supported by the presence of several potential METTL14 regulatory signature sequences in the sequence surrounding miR-133b (Fig 2, 5).

Some of these relationships have already been demonstrated in model organisms, most notably the role for both METTL3/METTL14 complex and miR-133b regulating tissue differentiation/proliferation, though in independent studies. However, the linking of miR-133b as a substrate of the METTL 3/14 complex, establishing a mechanism for this known developmental programming switch, and ex vivo evaluation of the above in a non-model system are all novel. The biggest weakness of the paper is the establishment of miR-133b. The results as presented certainly demonstrate a relationship between METTL14 and miR133b, and correlate miR-133b expression with tissue proliferation, but a functional or causative relationship between miR-133b and the proliferative status of tissue/genes of interest was not investigated. Establishing a specific mechanism for METTL14-mediated transition to musculature differentiation, particularly in a novel organism and with novel target miR-133b would be an interesting additional exploration.

Detailed comments

Major:

1. METTL14 and miR-133b do have putatively relevant function and the authors demonstrate correlated expression levels, but this is not functionally or causatively investigated with regard to the role of miR-133b muscle cell proliferation. A phenotype for overexpressing or inhibiting the regulatory activity of miR-133b, without the confounding factor of METTL14 over/underexpression, would much more strongly support the title and some conclusions of the manuscript. Could the authors transfect synthetic/isolated miR-133b utilizing their already established lentiviral or lipofectamine system?

Response: Thank you for your insightful comments. We appreciate your suggestion regarding the direct investigation of miR-133b's role in muscle cell proliferation. While our current findings suggest that METTL14 influences miR-133b expression, previous studies have indeed shown that miR-133b plays a role in muscle cell proliferation and differentiation. However, we recognize that we lack direct evidence for this effect in ducks, which are not a typical model organism. In response to this, we have planned additional experiments using miRNA mimics and inhibitors to further explore the impact of miR-133b on muscle proliferation and differentiation in ducks. These experiments are currently in progress, and we intend to include the results in a future manuscript that will focus specifically on the maturation mechanism of miRNA in the proliferation and differentiation of myoblasts in duck pectoral muscle.

2. Genes assessed in figure 1 are used as markers for developmental stage and/or evidence of cellular processes without clarification or discussion (example – line 285/290). Elaborating on each slightly would much more strongly support their role in the manuscript, particularly those not assessed in the reference establishing E19 as a developmental transition - CDK2, CyclinD1, and MYHC.

Response: Thank you for your valuable suggestion. We agree that providing further clarification on the marker genes would help readers better understand the rationale behind their selection. As a result, we have added a detailed explanation of the selection criteria for CDK2, CyclinD1, MYOG, and MYHC in lines 309-324. This additional information will better support the role of these genes in the manuscript.

3. In lines 307-316 it’s unclear how or why these six particular miRNAs were selected for analysis. The text mentions previous studies, but none are cited.

Response: Thank you for your comment. I have now added references to previous studies to support the selection of these six miRNAs for analysis. This should help clarify the rationale behind their inclusion in our study.

4. A few statistical analyses/pieces of information are not readily visible on figures. Differences in m6a methylation are described as though significant, but significance is not shown on Fig2G. Figures 3 and 4D similarly appear to show differences in differentiation gene expression and proliferative rate respectively, but the results of statistical analysis are not shown. Lastly, it appears as though the highlighted regions of figure 5A are METTL14 m6a modifying sequences, but this is not mentioned in the legend/text and the sequences in the figure are somewhat difficult to read.

Response: Thank you for your valuable comments. Regarding the differences in m6A methylation, we did not intend to describe them as statistically significant but rather as varying across different stages. As such, Fig 2G does not display statistical significance. We also acknowledge that statistical analysis results were not included in Figures 3 and 4D, and we have since updated the figures to reflect this. Regarding Figure 5A, we would like to clarify that the highlighted regions correspond to the predicted pri-miR-133b sequence rather than METTL14 m6A modifying sequences. The motifs indicated are potential m6A modification sites. To provide clearer context, we have revised the description to: "Therefore, this study proposes predicting the pri-miRNA sequence by taking the position of pre-miRNA in the genome and extending 500 base pairs before and after it (Fig 5A)." We hope this adjustment helps to clarify the figure and its context.

5.Similarly, I was not able to determine sample sizes throughout.

Response: Thank you for your comment. All fluorescence quantitative images in the manuscript were obtained using three biological replicates, with each sample measured in triplicate. We have now included a description of the data analysis process (2.14) in the manuscript for better clarity.

Minor

1. The acronym ‘m6a’ is not formally defined.

Response: Thank you for pointing that out. We have added the formal definition of ‘m6A’ in the first paragraph, line 57 of the manuscript.

2.A few conjugation/sp/grammatical errors, e.g.: line 119 “our previously research”, the timeline of sample collection on line 139 is somewhat unclear in “eggs were selected… post-hatch” (does this mean individuals?), line 142 “stored at -80 for further.”, line 318 “In the previous research” should perhaps be “in previous studies”, line 349 “conducted the detection” should just be “detected”, line 471 “apply to all animals” should perhaps be “is conserved in this system” to maintain scope, the sentence “And want to further verify..” on line 519-520 is an incomplete sentence.

Response: Thank you for pointing out these grammatical errors. We have made the following revisions to address your concerns:

“our previously research” has been changed to "our research revealed."

“eggs were selected… post-hatch” has been revised to "eggs were selected on embryonic days 13 (E13), 19 (E19), and 27 (E27)."

“stored at -80 for further” has been revised to "stored at -80°C for further analysis."

“In the previous research” has been changed to "in previous studies."

“conducted the detection” has been revised to "detected."

“apply to all animals” has been revised to "is conserved in this system."

“And want to further verify...” has been corrected to "Further validation is required to determine which of these miRNAs are regulated by METTL14."

3. Some reorganizations could help significantly with regards to reader comprehension as well as supporting the authors’ argument. While the manuscript as presented does follow a logical ordering, moving from in vivo proof of concept to more causative in vitro experiments, some sections feel out of place. Most notably sections 3.5 and 3.6 disrupt the ideological story by pivoting rapidly from m6a modified RNAs in vivo, back to the developmental status marker genes in sections 3.1 and 3.2, and then return to the change in expression of specific miRNAs in the newly established in vitro system.

Response: Thank you for your comment. In this manuscript, sections 3.1-3.4 describe in vivo experiments, while sections 3.5-3.8 focus on in vitro experiments. Sections 3.1, 3.2, 3.5, and 3.6 present marker gene analysis at different time points and stages. To improve the flow and logical structure of the manuscript, I have added transitional sentences in section 3.4 to provide better coherence.

4. The differentiation media used to establish the ex vivo system in line 340 is not specified in the materials and methods or cited in references.

Response: Thank you for pointing that out. We have now included the detailed differentiation medium formulation in section 2.3, "Isolation, culture, and induction of differentiation of duck embryonic pectoral muscle myoblasts," to provide clarity.

5. The first few sentences of 3.7 are redundant with the well described methods section. Of more use here would be the contents of and purpose for the plasmids transfected.

Response: Thank you for your suggestion. We have revised 3.7 section by removing the redundant content, making the results more concise and focused on the contents and purpose of the plasmids transfected.

6. In lines 375-377 the conclusions of the proliferation assessment are offered before the data. This sentence should instead be centered on the purpose and hypothesis for these data. In addition, the proliferative markers assessed in lines 382-390 are more consistent with the preliminary experiments overall. Flipping these two sets of data would allow for increasingly granular assessment of proliferation.

Response: Thank you for your suggestion. We have swapped the order of the data and analysis, first presenting the data followed by the analysis, to provide a more coherent and detailed assessment of proliferation in line 406-410.

7. Some information that, in the introduction or alongside data, would support the relationship between METTL14 and miR-133b is instead in the discussion. E.g. the discussion in lines 445-447 establishing E19 as a critical turning point in duck pectoral muscle, or the established regulatory role for miR-133b in differentiation/proliferation of other muscle tissue in other systems in 521-523.

Response: Thank you for your valuable feedback. We have revised the manuscript to include the developmental patterns of duck pectoral muscle and the important role of miR-133b in regulating skeletal muscle differentiation and proliferation in the Introduction section. We hope this enhances the clarity and flow of the manuscript.

8. There are a few places when summarizing data in the discussion, e.g. references to the expression of MYOG and MYHC in 450-453, that would benefit from directing the reader back to the relevant figure (here, fig. 1).

Response: Thank you for your suggestion. We have added references to the relevant figures, including Fig. 1, in the Discussion section to direct the reader back to the corresponding data.

9.In lines 481-486, the authors present a proof-of-concept for the METTL3/14 complex in RNA regulation, but it is unclear the extent to which each METTL protein is acting. Is there a reason the authors focused only on METTL14?

Response: Thank you for your insightful comment. In our preliminary study, MeRIP-seq and miRNA-seq analyses revealed significant differences in the expression of several m6A-related genes across three developmental stages. Among these, METTL14, an m6A writer, showed notable variations, as referenced in citation 21 of this manuscript. However, METTL3 was not enriched in the results, which might be due to sequencing limitations or the incomplete annotation of the duck genome. Therefore, in this study, we chose to focus on the impact of METTL14 for further investigation.

Reviewer #2: In the current manuscript, “METTL14 Regulates Proliferation and Differentiation of Duck Myoblasts Through Targeting MiR-133b” the authors have shown the importance of METTL14, which is a core component of the m6A methylation transferase complex, in the development of Duck pectoral muscle. Further, authors have also shown that the overexpression of METTL14 increased the expression of miR-133b, whose precursor sequence contains m6A modification sites, suggesting that METTL14 may participate in the regulation of muscle development by affecting the expression of miR-133b. Overall this study provides a new perspective of the molecular mechanisms involved in the development of duck pectoral muscle. This research appears to have therapeutic significance since it offers potential molecular targets for the genetic improvement of duck pectoral muscle. I recommend the manuscript for publication subject to responses to the following comments

Minor comments

1)Authors should consider adding individual data points in all of the bar graphs presented in result sections.

Response: Thank you for your suggestion. We have added individual data points to the bar graphs presented in the Results sections.

2)Authors should consider making a graphical abstract to summarise their findings.

Response: Thank you for your suggestion. We have created a graphical abstract to summarize our findings and have included it in the manuscript.

3) Authors are claiming that the overexpression of METTL14 increased the expression of miR-133b, whose precursor sequence contains m6A modification sites, suggesting that METTL14 may participate in the regulation of muscle development by affecting the expression of miR-133b. It would be interesting if authors could also show the effect of miR-133b mimic and inhibitor on pectoral muscle development and its effect on expression of METTL4.

Response: Thank you for your valuable suggestion. We agree that further experiments using miR-133b mimics and inhibitors are necessary to better understand its role in muscle development and whether it is regulated through m6A-modified motifs. We are currently conducting these experiments, and the results will be incorporated into a separate manuscript focusing on the effects of miRNA on the proliferation and differentiation of duck pectoral muscle myoblasts.

---

## [Decision Letter · Decision Letter 1]

24 Feb 2025

METTL14 Regulates Proliferation and Differentiation of Duck Myoblasts Through Targeting MiR-133b

PONE-D-24-38664R1

Dear Dr. Gu,

We’re pleased to inform you that your manuscript has been judged scientifically suitable for publication and will be formally accepted for publication once it meets all outstanding technical requirements.

Kind regards,

Abdul Qadir Syed, PhD

Academic Editor

PLOS ONE

Reviewers' comments:

Reviewer's Responses to Questions

**Comments to the Author**

1. If the authors have adequately addressed your comments raised in a previous round of review and you feel that this manuscript is now acceptable for publication, you may indicate that here to bypass the “Comments to the Author” section, enter your conflict of interest statement in the “Confidential to Editor” section, and submit your "Accept" recommendation.

Reviewer #1: All comments have been addressed

Reviewer #2: All comments have been addressed

2. Is the manuscript technically sound, and do the data support the conclusions?

Reviewer #1: Yes

Reviewer #2: Yes

3. Has the statistical analysis been performed appropriately and rigorously? 

Reviewer #1: Yes

Reviewer #2: Yes

4. Have the authors made all data underlying the findings in their manuscript fully available?

Reviewer #1: Yes

Reviewer #2: Yes

5. Is the manuscript presented in an intelligible fashion and written in standard English?

Reviewer #1: Yes

Reviewer #2: Yes

6. Review Comments to the Author

Reviewer #1: All of my comments were addressed, and the manuscript is far more clear. I have only two remaining comments, both of which are relatively minor or unnecessary copyediting.

1) My question about the interpretation of the statistical analysis in Figure 2G was responded to, but still seems somewhat ambiguous. I don't think it is fair to allude to the presence of a difference if that difference is not borne out in the statistical analysis of the data. That said, the authors' point that the pattern of m6a modification is non-identical at these timepoints may not require such rigor, or may be demonstrated by using a less specific statistical analysis.

2) Some of the materials and methods section is currently written in an instructive present tense rather than past tense (e.g. line 261 - "Revive low-passage 293T cells in advance and culture them using DMEM medium 261

containing 10% FBS and 1% Penicillin Streptomycin." vs line 150 "Duck embryos were obtained from Hainan Chuanwei Peking Duck Farming Co., Ltd.,...").

Otherwise, all revisions addressed my comments satisfactorily. I look forward to the followup studies examining the mechanism of METTL14 and miR133b on muscle development!

Reviewer #2: Authors have addressed all the comments and modified the manuscript can now be accepted in its current form.

7. PLOS authors have the option to publish the peer review history of their article (what does this mean? ). If published, this will include your full peer review and any attached files.

**Do you want your identity to be public for this peer review?** For information about this choice, including consent withdrawal, please see our Privacy Policy .

Reviewer #1: **Yes: ** Joshua W Thompson

Reviewer #2: No

---

## [Editor Report · Acceptance letter]

PONE-D-24-38664R1

PLOS ONE

Dear Dr. Gu,

I'm pleased to inform you that your manuscript has been deemed suitable for publication in PLOS ONE. Congratulations! Your manuscript is now being handed over to our production team.

Kind regards,

on behalf of

Dr. Abdul Qadir Syed

Academic Editor

PLOS ONE